# BMP-2 Genome-Edited Human MSCs Protect against Cartilage Degeneration via Suppression of IL-34 in Collagen-Induced Arthritis

**DOI:** 10.3390/ijms24098223

**Published:** 2023-05-04

**Authors:** Dong-Sik Chae, Seongho Han, Min-Kyung Lee, Sung-Whan Kim

**Affiliations:** 1Department of Orthopedic Surgery, Catholic Kwandong University College of Medicine, International St. Mary’s Hospital, Incheon 22711, Republic of Korea; 2Department of Family Medicine, Dong-A University College of Medicine, Dong-A University Medical Center, Busan 49201, Republic of Korea; 3Department of Dental Hygiene, Dong-Eui University, Busan 47340, Republic of Korea; 4Department Medicine, Catholic Kwandong University College of Medicine, Gangneung 25601, Republic of Korea

**Keywords:** BMP-2, collagen-induced arthritis, genome editing, IL-34, mesenchymal stem cells

## Abstract

Even though the regenerative potential of mesenchymal stem cells (MSCs) has been extensively studied, there is a debate regarding their minimal therapeutic properties. Bone morphogenetic proteins (BMP) are involved in cartilage metabolism, chondrogenesis, and bone healing. In this study, we aimed to analyze the role of genome-edited BMP-2 overexpressing amniotic mesenchymal stem cells (AMMs) in a mouse model of collagen-induced arthritis (CIA). The BMP-2 gene was synthesized and inserted into AMMs using transcription activator-like effector nucleases (TALENs), and BMP-2-overexpressing AMMs (AMM/B) were sorted and characterized using quantitative reverse transcription polymerase chain reaction (qRT-PCR). The co-culture of AMM/B with tumor necrosis factor (TNF)-α-treated synovial fibroblasts significantly decreased the levels of interleukin (IL)-34. The therapeutic properties of AMM/B were evaluated using the CIA mouse model. The injection of AMM/B attenuated CIA progression and inhibited T helper (Th)17 cell activation in CIA mice. In addition, the AMM/B injection increased proteoglycan expression in cartilage and decreased the infiltration of inflammatory cells and factors, including IL-1β, TNF-α, cyclooxygenase (COX)-2, and Nuclear factor kappa B (NF-kB) in the joint tissues. Therefore, editing the BMP-2 genome in MSCs might be an alternative strategy to enhance their therapeutic potential for treating cartilage degeneration in arthritic joints.

## 1. Introduction 

Arthritis is a debilitating disease affecting millions of people worldwide [1]. Despite the availability of several treatment options, the disease remains incurable. Although autologous chondrocyte implantation has been used to treat cartilage lesions, it has some limitations, including a low recovery rate and invasiveness [2]. Other therapeutic approaches primarily aim to alleviate disease symptoms without effectively reducing the deterioration of tissue structures. 

The inflammatory microenvironment resulting from articular cartilage injury plays a significant role in chondrocyte hypertrophy, extracellular matrix degradation, bone formation, and the development of osteoarthritis [3]. Synovial inflammation is also associated with knee dysfunction and contributes to the occurrence and progression of osteoarthritis [4]. Furthermore, immune cells not only suppress inflammation but also facilitate tissue repair [5]. To repair articular cartilage damage, it is important to improve the regenerative microenvironment of the joint, which involves enhancing the recruitment of endogenous stem cells, regulating local immunity, and protecting chondrocytes and their matrix. 

Stem-cell-based tissue engineering has recently emerged as a promising tool for repairing damaged cartilage. Mesenchymal stem cells (MSCs) derived from various tissues, including adipose, dental pulp, bone marrow, umbilical cord, and placenta, are promising alternatives to chondrocytes due to their differentiation potential [6]. MSC transplantation promote the regeneration of injured cartilage via three distinct mechanisms: differentiation into chondrocytes, secretion of growth factors, and suppression of inflammation [7,8]. The paracrine products of MSCs can promote cell-to-cell communication and play a significant role in mediating tissue repair and regeneration. However, there is ongoing debate regarding the minimal therapeutic properties of MSCs for cartilage repair [9], emphasizing the need for further research in this area.

The chondrogenic differentiation of MSCs requires morphological transition growth factors, such as transforming growth factor (TGF)-β1, -β2, -β3, and bone morphogenetic protein (BMP) [10,11]. BMP promotes the chondrogenic effects of TGF-β [12]. Specifically, BMP-2 is chondrogenic in vitro and in vivo [13] and stimulates synovial chondrogenesis [14]. Recent research demonstrates that inhibiting SMAD7 with microRNA-18c-5p increases BMP-2 levels in MSCs, facilitating the repair of cartilage injury [15]. Furthermore, the injection of MSCs with adenovirus-transduced BMP-2 into an articular fracture model has been shown to result in the effective repair of both bone and cartilage [13]. In a mouse femur model, lentiviral BMP-2 transduced MSCs stimulated host cells to differentiate into an osteoblastic lineage [16]. However, the therapeutic mechanisms of these compounds in damaged cartilage have not been fully elucidated. 

This study aimed to investigate the characteristics and therapeutic mechanism of genome-edited BMP-2 overexpressing human amniotic MSCs (AMM/B) in a collagen-induced arthritis model. Our findings suggest that AMM/B has a positive immunomodulatory effect and may improve cartilage repair or alleviate joint arthritis. 

## 2. Results

### 2.1. Targeted Knock-in of BMP-2

To create a stable AMM cell line expressing BMP-2 via gene editing, we used transcription-activator-like effector nucleases (TALENs). The donor plasmid used for targeting contained BMP-2 and GFP-T2A-Puromycin under the control of the phosphoglycerate kinase (PGK) promoter and elongation factor-1 alpha (EF1α) promoter, respectively, and was designed to be integrated into the AAVS1 site on chromosome 19 (Figure 1A). The AMMs were transfected with a donor plasmid and a pair of TALENs. The transfected AMMs were isolated by culturing with puromycin, resulting in less than 10% GFP-positive cells. The GFP-positive cells were isolated using fluorescence-activated cell sorting (FACS), yielding 98.2% GFP-positive cells (Figure 1B). To confirm the successful integration of the donor plasmid into the AAVS1 site, the genomic DNA was subjected to PCR followed by touchdown PCR. The correct integration of the donor plasmid was confirmed by amplifying the 5′ junction fragment (960 bp) (Figure 1C). Finally, qRT-PCR was used to confirm the BMP-2 expression in the transfected AMMs, and the BMP-2 gene and protein levels were found to be significantly increased in the AMM/B cell line compared to normal AMMs (Figure 1D,E). The resulting stable AMM cell line expressing BMP-2 (AMM/B) was used in this study.

### 2.2. In Vitro Immunomodulatory Potential of AMM/B

To investigate the immunomodulatory effects of AMM/B on synovial fibroblasts in vitro, the synovial fibroblasts were treated with or without tumor necrosis factor (TNF)-α and co-cultured with AMM/B and AMMs. The cytokine analysis was performed using supernatants from two-day co-cultures. The enzyme-linked immunosorbent assay (ELISA) results revealed that the co-culture with AMM/B significantly decreased the IL-34 levels compared to co-culturing with AMMs (Figure 2). 

### 2.3. Anti-Arthritogenic Potential of AMM/B in a Collagen-Induced Arthritis (CIA) Mouse Model 

The CIA mice were administered AMM/B, PBS, or AMMs by injection to evaluate the anti-arthritic properties of AMM/B in damaged cartilage in vivo (Figure 3A). Interestingly, we found that AMM/B administration significantly decreased the arthritis clinical score at 5, 10, and 15 days compared with PBS or AMMs injection (Figure 3B,C).

To further investigate the mechanisms responsible for the anti-arthritogenic effects of AMM/B, we evaluated the effect of AMM/B administration on T cells using flow cytometry. Compared to PBS or AMM administration, AMM/B administration significantly decreased the number of Th17 cells (Figure 4A,B). We also measured the concentration of IL-17A in mouse serum and found that the AMM/B-administered group had significantly lower levels of IL-17A than the PBS or AMM-administered groups (Figure 4C).

### 2.4. Histological Analysis after AMM/B Injection 

We stained joint tissue sections with Safranin O to assess the cartilage-protective effects of AMM/B. Our results showed that AMM/B administration increased the proteoglycan expression in cartilage compared to the PBS or AMM administration, indicating protection against cartilage degradation (Figure 5A,B). In addition, hematoxylin and eosin staining was performed to evaluate the anti-inflammatory response. The histological analysis revealed that, compared to PBS or AMM, the administration of AMM/B significantly reduced inflammatory cell infiltration (Figure 5C,D).

### 2.5. Gene Expression Analysis in Joints after AMM/B Injection

After cell transplantation, we analyzed the levels of pro-inflammatory factors in joint tissues to further investigate the anti-arthritic therapeutic mechanisms. Notably, the levels of pro-inflammatory factors IL-1β, TNF-α, COX-2, and NF-kB were significantly decreased in the AMM/B-administered joint tissues compared to that in the PBS- or AMM-administered joint tissues (Figure 6).

## 3. Discussion

This study explored the therapeutic properties and mechanisms of action of BMP-2 genome-edited AMMs in a CIA mice model. Our primary findings were as follows: (1) AMM/B-secreted factors showed anti-inflammatory properties in vitro; (2) AMM/B transplantation attenuated CIA progression; and (3) AMM/B transplantation reduced the number of Th17 cells and the levels of pro-inflammatory factors in CIA. These findings suggest that BMP-2 genome-edited AMMs are promising alternatives for cartilage repair or joint arthritis prevention. 

Although MSCs have been shown to mitigate osteoarthritis progression and have the potential for cartilage repair, their therapeutic potential remains controversial owing to their low efficacy [17,18]. The primary limitation of this therapeutic approach is the low survival rate of transplanted cells (0.2–10%) [19]. A significant number of transplanted cells undergo cell death in the short term, which may be attributed to environmental stress after engraftment. Recent clinical trials on MSC-based cartilage regeneration have not yielded sufficient evidence for long-term restoration of the original hyaline cartilage and improvement of osteoarthritis [9]. Despite controversies surrounding the therapeutic effects of stem cells, stem cell therapy remains an attractive technique. Treatment outcomes can be improved by initiating therapeutic interventions during the early stages of joint inflammation [20]. Therefore, we aimed to enhance the therapeutic potential of MSCs by generating a genetically modified cell line using genome-editing technology. This approach enables the targeting of specific and safe genomic sites to minimize genetic mutagenesis. 

BMP-2 is a member of the TGF-β superfamily and is primarily involved in regulating bone growth and development. BMP-2 signaling has been demonstrated to facilitate healing by stimulating osteoblast differentiation and bone regeneration. Several studies have reported that BMP-2 can induce the chondrogenic differentiation of mesenchymal stem cells (MSCs) [21]. Moreover, BMP-2 promotes cartilage growth by inducing extracellular matrix synthesis [22]. Recombinant human (rh) BMP-2 injection induces proteoglycan synthesis in joints [23], and BMP-2 implantation with biomaterials enhances regeneration in damaged cartilage [24]. Accumulating evidence suggests that BMP-2 can restore synovial joint function. Therefore, in this study, we hypothesized that BMP-2 overexpression in MSCs could enhance cartilage repair and protect against joint arthritis.

IL-34, a cytokine predominantly expressed in macrophages, osteoclasts, and synovial fibroblasts of RA patients, acts as a downstream effector of IL-1β and TNF-α and is recognized to be involved in the pathogenesis of RA [25,26,27]. It is critical in regulating immune cell function, bone remodeling, and repair. The expression of IL-34 is closely associated with inflammation, leukocyte count, and severity of synovitis in patients [27]. To repair the cartilage in arthritic joints, inflammatory cytokines must be downregulated, and cartilage-producing cells must be introduced. BMP-2 inhibits the expression of IL-34 in the synovial fibroblasts of RA patients via ALK1 and ALK5 receptors [28]. Consistent with this report TNF-α-treated synovial fibroblasts exhibited high expression of IL-34. However, co-culturing synovial fibroblasts with AMM/B decreased IL-34 expression. Moreover, the expression levels of the pro-inflammatory cytokines, IL-1β and TNF-α, were decreased in the AMM/B-injected tissues.

CD4+ T cells can differentiate into various subsets, including regulatory T (Treg), T helper type 1 (Th1), Th2, and Th17 cells. Th17 cells play a significant role in RA, and the percentage of Th17 cells in RA patients is higher than in healthy individuals [29]. Recent reports indicate that IL-34 stimulates the proliferation and differentiation of Th17 cells [30]. These findings are consistent with our observations that the number of Th17 cells increased in arthritic mice. In addition, AMM/B-treated CIA mice exhibited a significant suppression of Th17 cells and were protected against cartilage damage. Thus, our results suggest that the therapeutic effect of BMP-2 secreting AMM in arthritic mice may be attributed to the regulation of Th17 cells via the inhibition of IL-34 expression.

Inflammatory cells release pro-inflammatory cytokines, which play crucial roles in synovial inflammation and cartilage degradation [31]. These cytokines upregulate the production of matrix-degrading enzymes and suppress proteoglycan synthesis, leading to joint destruction [32]. High levels of representative pro-inflammatory cytokines, IL-1β and TNF-α, have been observed in the synovium of patients with osteoarthritis [33]. Neutralizing IL-1β or TNF-α has been proven to be a successful treatment strategy for RA. Previous studies have reported the immunomodulatory role of BMP-2 in macrophages [34]. When supplemented with BMP-2, the expression of M1 phenotypic markers, such as IL-1β, IL-6, and iNOS, was significantly reduced in M1 polarized macrophages [34]. This suggests that BMP-2 has immunoregulatory properties in inflammatory conditions. Therefore, we investigated the expression of inflammatory cytokines in CIA joint tissues following cell injection. Interestingly, AMM/B injection significantly reduced the levels of the inflammatory cytokines IL-1β and TNF-α and inflammatory cell infiltration compared to AMM injection alone. These findings suggest that BMP-2 might have a role in reducing cytokine levels, potentially by directly affecting the cells producing the cytokines (autonomous effect). In addition, the concurrent use of AMMs and BMP-2 synergistically modulates the immune response in CIA-affected joint tissues. Previous studies have demonstrated that untreated MSCs can reduce the levels of pro-inflammatory cytokines such as IL-1β, IL-6, and TNF-α and mitigate chondrocyte apoptosis [35]. Furthermore, AMMs possess anti-inflammatory, chondroprotective, and regenerative properties [36,37]. 

## 4. Material and Methods

### 4.1. Cell Culture 

The study used human amniotic mesenchymal stem cells (AMMs) obtained from Thermo Scientific, Inc. (Waltham, MA, USA). The cells were cultured in low-glucose Dulbecco’s Modified Eagle Medium (DMEM; GIBCO, Grans Island, NY, USA) with 100 U/mL penicillin, 100 µg/mL streptomycin (GIBCO), and 10% fetal bovine serum (FBS) [38]. The culture media were changed every three days, and, when the cells reached confluence after 1–2 weeks, they were passaged using trypsinization. For all subsequent assays, the cells that had been passaged less than 5 times were used. To detach the cells, 0.05% trypsin-EDTA (Gibco) was used, and the cells were incubated for 20 min at 37 °C.

### 4.2. Donor Vector Construction

To construct the donor vector, BMP-2 was synthesized and inserted into the adeno-associated virus integration site 1 (AAVS1) safe harbor site, targeting the donor vector (System Biosciences, Palo Alto, CA, USA) at the Nde and SalⅠrestriction sites [38]. This donor vector carries an expression cassette that contains the PGK promoter-driven BMP-2 and EF1α promoter-driven GFP-T2A-puromycin (Figure 1A).

### 4.3. Transfection and Drug Selection 

For electroporation, the AMMs (1 × 10^5^ cells) were resuspended with AAVS1 left TALE-Nuclease vector (System Biosciences), AAVS1 right TALE-Nuclease vector (System Biosciences), and BMP-2 (AAVS1) HR Donor (System Biosciences) in electroporation buffer [39]. The cells and plasmids were mixed inside the Neon pipette tip and were electroporated using the Neon Transfection System (Thermo Fisher Scientific, Whadam, MA, USA). Additionally, the BMP-2 knock-in AMMs were cultured for 10 days and incubated with 5 μg/mL puromycin for 1 week. Green fluorescent protein (GFP) expressing AMM/B were observed using a florescent microscope. 

### 4.4. Fluorescence-Activated Cell Sorting (FACS)

To isolate only AMM/B, FACS was performed as previously described [40]. Briefly, the puromycin-selected AMMs grew for 30 days. The presence or absence of GFP expression was analyzed by observing the cells under the microscope. The puromycin-selected cells were washed with phosphate-buffered saline (PBS) to remove any residual media or contaminants. Next, the cells were detached by treating them with 0.05% Trypsin/EDTA, a solution that breaks down proteins and chelates metal ions to aid in cell detachment. Following detachment, the cells were resuspended in PBS to ensure a uniform suspension. Finally, the GFP-expressing AMM/B were sorted using the S3e Cell Sorter (Bio-Rad in Hercules, CA, USA), resulting in a population of cells where 98.2% of the cells were GFP-positive. The sorted AMM/B were cultured for 4 weeks for the next experiments. In this study, we used the AMM/B that had undergone fewer than six passages.

### 4.5. Genomic DNA Extraction and Junction PCR

The genomic DNA from AMMs or AMM/B was extracted using a G-spin™ Total DNA Extraction Mini Kit (Intron Biotechnology, Seongnam, Republic of Korea). Next, touch-down PCR was used to amplify 120 ng of genomic DNA, consisting of 36 cycles [39]. The touch-down PCR conditions were as follows: one cycle at 98 °C for 30 s, followed by 22 cycles of 98 °C for 30 s, 72–60 °C for 30 s (with a decrease of 1 °C every two cycles), and 72 °C for 1 min. This was followed by 14 additional cycles at 98 °C for 30 s, 60 °C for 30 s, and 72 °C for 1 min, and the final extension step at 72 °C for 10 min. For the second PCR, 0.5 μL of the touch-down PCR product was used. The second PCR conditions were as follows: one cycle at 98 °C for 30 s, followed by 35 cycles at 98 °C for 30 s, 65 °C for 30 s, and 72 °C for 1 min, with a final extension step at 72 °C for 10 min [39]. 

### 4.6. Quantitative Reverse Transcription PCR (qRT-PCR) 

The experimental procedures for the qRT-PCR assays were conducted following previously reported methods [41,42]. In brief, the total RNA was extracted from cells using RNA-stat (Iso-Tex Diagnostics, Friendswood, TX, USA), and the extracted RNA was reverse transcribed using TaqMan reagents (Applied Biosystems, Foster City, CA, USA). The resulting cDNA was subjected to qRT-PCR using specific primers and probes. The RNA levels were quantified using the ABI PRISM 7000 instrument (Applied Biosystems), and the relative mRNA level was normalized to the expression of GAPDH. The gene expression data were analyzed using the formula Rel Exp = 2^−ΔCT^ (fold difference), where ΔCT = (Ct of target gene) − (Ct of endogenous control gene, GAPDH) for the experimental samples. The relative expression value of the target gene was normalized to GAPDH and expressed relative to a calibrator. The number of PCR cycles was determined using the software of the ABI Prism sequence detection system. The qRT-PCR primers used in this study were as follows: human BMP-2 (Hs00154192_m1) and GAPDH (Hs99999905_m1), and mouse IL-1β (Mm00434228_m1), TNF-α (Mm00443258_m1), COX2 (Mm03294838_g1), NF-kB (Mm00476361_m1), and GAPDH (Mm99999915_g1). All the primers and probes were purchased from Applied Biosystems.

### 4.7. Isolation of Synovial Fibroblast 

The mouse synovial fibroblasts were isolated from the synovial tissue of DBA/1 mice (OrientBio, Seongnam, Republic of Korea) [43]. In brief, the synovial tissue was finely minced and filtered using sterile 100 μm nylon filter (BD Biosciences, San Jose, CA, USA). The isolated cells were cultured in Dulbecco’s modified minimal essential medium (F-15) media. The isolated cells were passaged several times prior to use. Two separate primary cell cultures of fibroblasts were analyzed, each derived from pooled synovium from 10 mice per preparation. The cells exhibited typical synovial fibroblast morphology and were confirmed to express vascular cell adhesion molecule (VCAM)-1 through immunohistochemistry testing.

### 4.8. Co-Culture with Synovial Fibroblast and Enzyme-Linked Immunosorbent Assay (ELISA)

To investigate the immunomodulatory effects of AMM/B on synovial fibroblast, 5 × 10^5^ synovial fibroblasts were treated without or with 10 ng/mL tumor necrosis factor (TNF)-α for 24 h and then co-cultured with 5 × 10^5^ AMMs or AMM/B in RPMI 1640 containing 10% FBS. The co-cultured supernatants were harvested after two days, and the IL34 cytokine levels were evaluated using a murine IL-34 ELISA kit (R&D Systems, Minneapolis, MN, USA). The concentration of IL-17A was measured using murine IL-17A ELISA kits (R&D Systems) and mouse serum. The BMP-2 protein level was also examined using a human BMP-2 ELISA kit (R&D Systems).

### 4.9. Induction of Arthritis Model and Cell Injection

The Bovine type II collagen (Chondrex, Redmond, WA, USA) was emulsified using a complete Freund’s adjuvant (Chondrex) containing 2 mg/mL heat-killed *Mycobacterium tuberculosis* [44]. The six-week-old male DBA/1 mice (OrientBio) received a primary immunization, followed by a boosting immunization on day 21 using the same concentration of Bovine type II collagen and incomplete Freund’s adjuvant (Chondrex) [44]. The injection was intradermally conducted at the base of the tail. The severity was measured for 28 days after first injection. The severity of the arthritis was observed and scored as determined by the hind paw swelling and clinical scoring [44]; 1 × 10^6^ of AMMs or AMM/B were resuspended in 50 μL of PBS. When the arthritis score reached 3 or more, resuspended AMMs or AMM/B or 50μL of PBS were injected intraperitoneally twice a week. 

### 4.10. Flow Cytometry Analysis

The populations of Th17 cells were examined using flow cytometry. The methodology for the procedure was previously outlined in [45]. Briefly, the cells were suspended in Dulbecco’s PBS and incubated with PE- or FITC-conjugated antibodies at 4 °C for 20 min. The phycoerythrin (PE)-conjugated rat anti-mouse CD4 (eBioscience, San Diego, CA, USA) and fluorescein isothiocyanate (FITC)-conjugated rat anti-mouseIL-17A (eBioscience) antibodies were used. The cells were fixed in a 2% paraformaldehyde solution prior to analysis using a flow cytometer (BD, San Jose, CA, USA). The analyses were performed using CellQuest software (BD).

### 4.11. Histology and Analysis

The mice were euthanized with CO_2_ gas, and the tissues were obtained by performing dissection. The limbs and paw were fixed overnight in 4% paraformaldehyde and decalcified [44]. The cartilage and paws were embedded in paraffin and sectioned at 10 µm. The specimen was stained using Hematoxylin and Eosin (H&E) staining and safranin O (Science cell) to analyze the inflammation and confirm the cartilage destruction of the CIA model. The degree of cartilage degradation was assessed using a scale ranging from 0 to 3, where a score of 0 indicated no loss of proteoglycans, and a score of 3 indicated complete loss of staining for the proteoglycans [46]. The pathological changes were evaluated based on the level of inflammation in the cartilage and bone destruction, as described previously [47], using the following grading system: 0 denoted normal synovium, 1 denoted synovial membrane hypertrophy and cell infiltrates, 2 denoted pannus and cartilage erosion, 3 denoted major erosion of cartilage and subchondral bone, and 4 denoted loss of joint integrity and ankylosis. 

### 4.12. Statistical Analysis

The statistical analysis was performed using SPSS v12.0 (SPSS, Inc., Chicago, IL, USA) [39]. The data are presented as mean ± SD. A Student’s *t*-test was used to compare the two groups, and ANOVA with Bonferroni’s multiple comparison test was used for multiple group comparisons [39]. The statistical significance was determined at *p* < 0.05.

## 5. Conclusions

In conclusion, BMP-2 insertion into AMMs generated through gene editing may be a promising alternative strategy for enhancing anti-arthritic potential. Additionally, AMM/B could be a promising source of stem cells for autologous chondrocyte replacement, demonstrating robust therapeutic outcomes. However, this study has some limitations. First, the fates of the transplanted cells were not monitored. Second, using a mouse model may not accurately replicate human disease. Thirdly, the autonomous or non-autonomous effects of BMP-2 on cytokine production and the anti-inflammatory therapeutic mechanism regarding the interaction with other immune cells needs to be studied. Future investigations should focus on tracking AMM/B in vivo, translating the current findings to human trials, and assessing the long-term safety and efficacy of AMM/B treatment for arthritis. 

## Figures and Tables

**Figure 1 ijms-24-08223-f001:**
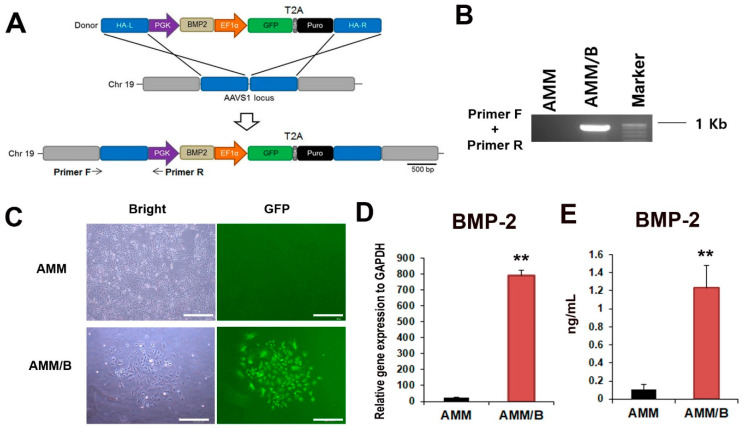
Generation of AMM/B using TALEN. (**A**) Schematic representation of the donor vector carrying the BMP-2 plasmid DNA. The expression cassette containing the PGK promoter-driven BMP-2 and EF1α promoter-driven GFP-T2A-puromycin was inserted into the AMMs genome via homology-directed repair (HR). The locations of primers are indicated (primers F and R). Abbreviations: PGK, phosphoglycerate kinase promoter; EF1α, elongation factor-1 alpha promoter; Puro, puromycin; HA-L, left homology arm; and HA-R, right homology arm. (**B**) The inserted donor plasmid was confirmed using junction PCR. (**C**) GFP-expressing AMM/B. Transfected cells were selected based on puromycin resistance, followed by FACS sorting. Bars = 500 μm. (**D**) Expression levels of BMP-2 were measured using qRT-PCR. ** *p* < 0.01, n = 4 each. (**E**) Protein expression levels of BMP-2 were measured using enzyme-linked immunosorbent assay (ELISA). ** *p* < 0.01, n = 4 each.

**Figure 2 ijms-24-08223-f002:**
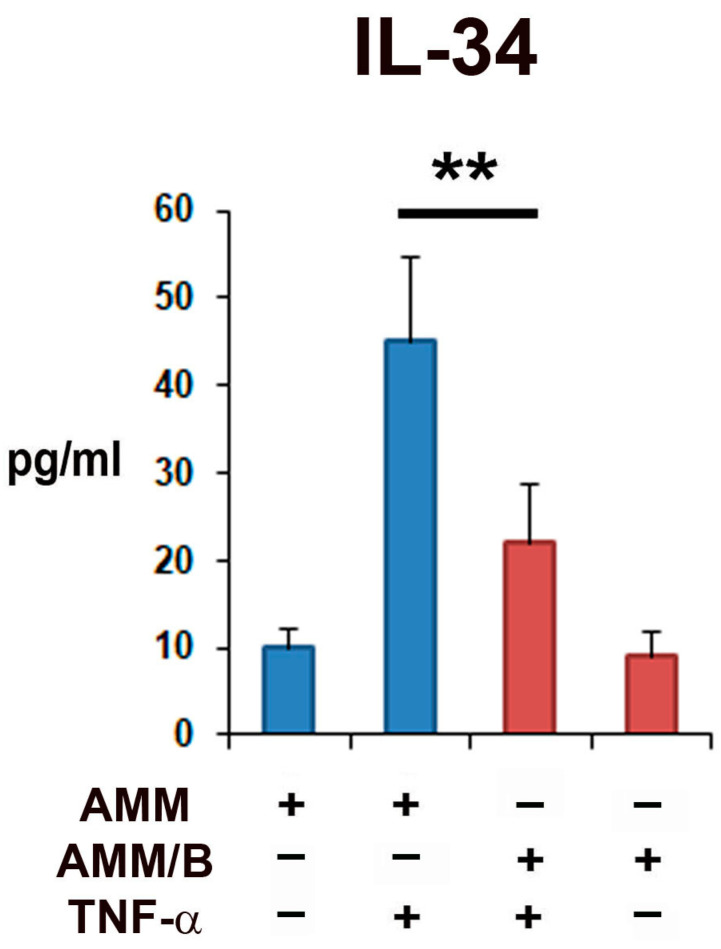
Anti-inflammatory properties of AMM/B. Synovial fibroblasts were co-cultured with AMMs or AMM/B after treatment with or without TNF-α. The concentration of IL-34 in supernatants was analyzed using ELISA. The ELISA assay results revealed that co-culturing with AMM/B significantly reduced IL-34 levels compared to co-culturing with AMMs. ** *p* < 0.01, n = 5 each.

**Figure 3 ijms-24-08223-f003:**
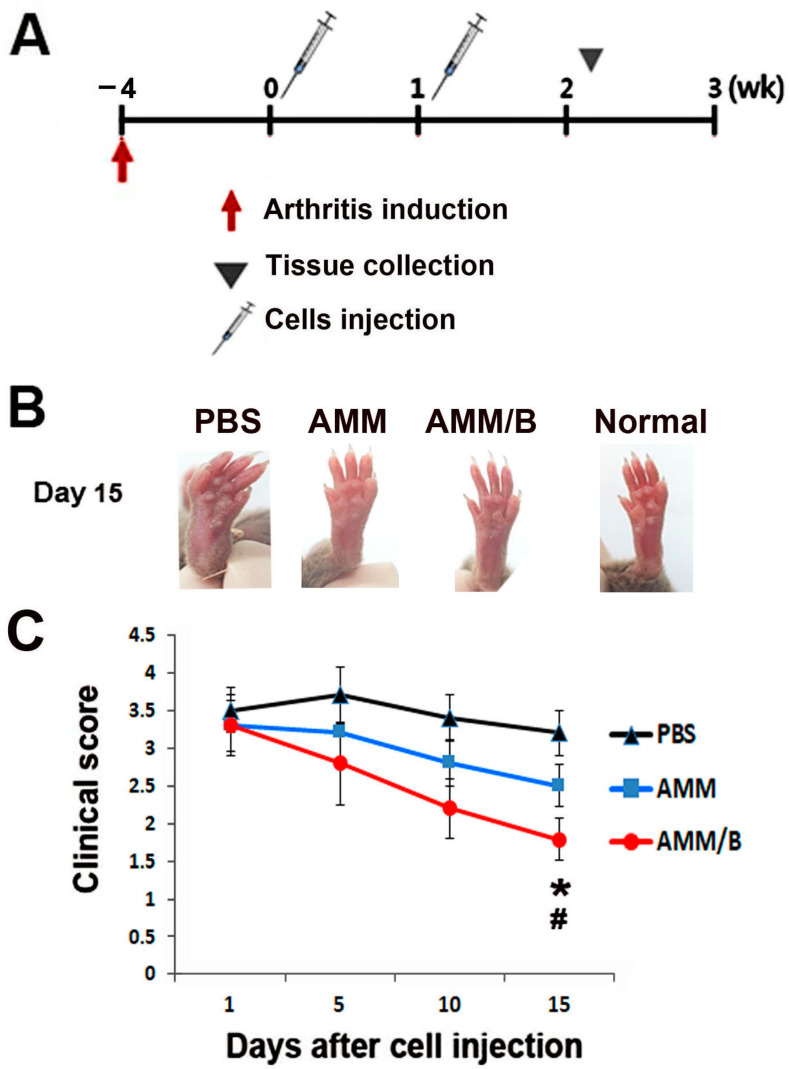
AMM/B injection protects against disease progression in CIA mice. (**A**) Schematic representation of the procedures for CIA induction, cell injection, and specimen harvest. (**B**) Representative photographs of mouse paws after cell transplantation. (**C**) Quantification of arthritis scores. Severe swelling was identified as a sign of arthritis. Injection of AMM/B led to a significant decrease in arthritis clinical scores at 5, 10, and 15 days compared to the injections of PBS or AMMs. # *p* < 0.01 AMM/B vs. PBS, * *p* < 0.05, AMM/B vs. AMM, n = 7 each.

**Figure 4 ijms-24-08223-f004:**
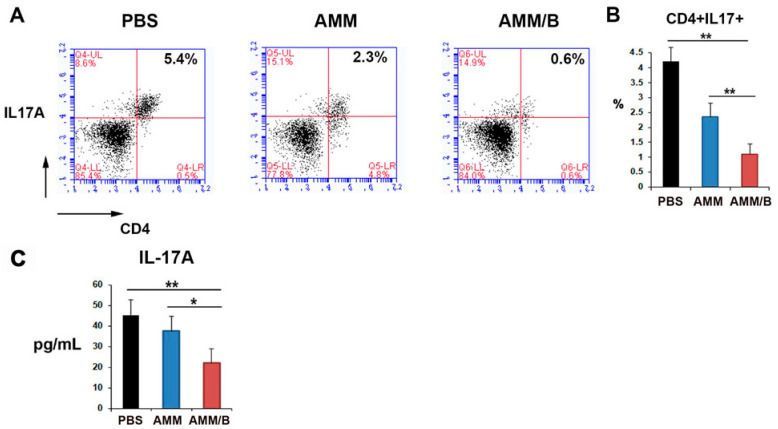
AMM/B injection influences Th17 cell populations in CIA mice. (**A**) Representative flow cytometry data for the detection of Th17 cells. (**B**) Quantitative data for Th17 cells were measured by analyzing blood samples from mice two weeks after cell injection. The number of Th17 cells decreased significantly following AMM/B injection compared to the PBS or AMMs injections. n = 7 each; ** *p* < 0.01. (**C**) Serum IL-17A concentration. Two weeks after cell transplantation, ELISA was performed using the serum of CIA mice. * *p* < 0.05,** *p* < 0.01, n = 7 each.

**Figure 5 ijms-24-08223-f005:**
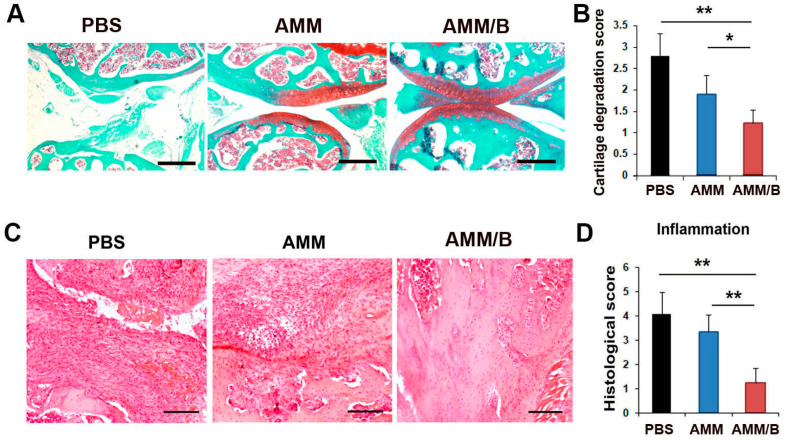
Histological analysis of joint tissue after injection of cells. (**A**) Representative photographs of Safranin O-stained sections. Proteoglycan expression was analyzed in the joint cartilage area of CIA mice after cell injection. Bars = 200 μm. (**B**) Quantification of cartilage degradation score. Loss of proteoglycans was examined using proteoglycan staining. n = 6 each; * *p* < 0.05 and ** *p* < 0.01. (**C**) Representative photographs of H&E-stained sections in the joint subchondral area. Bars = 200 μm. (**D**) Quantification of the inflammatory pathological score. Inflammatory pathological scores, such as mononuclear infiltration, were examined in the joint tissues. ** *p* < 0.01, n = 6 each.

**Figure 6 ijms-24-08223-f006:**
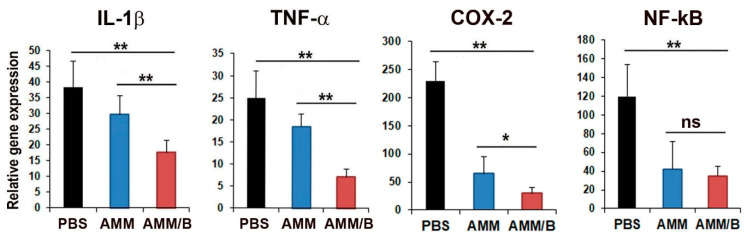
AMM/B injection suppresses inflammation in joint tissue. qRT-PCR analysis was performed on joint tissues injected with PBS, AMM, and AMM/B. AMM/B injection reduced the expression of inflammatory factors. The levels of pro-inflammatory factors IL-1β, TNF-α, COX-2, and NF-kB were significantly decreased in AMM/B-injected joint tissues. * *p* < 0.05, ** *p* < 0.01, n = 6 each. Abbreviations: ns, not significant.

## Data Availability

The data presented in this study are available on request from the corresponding author upon reasonable request.

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
