# Peer review of "BMP-2 Genome-Edited Human MSCs Protect against Cartilage Degeneration via Suppression of IL-34 in Collagen-Induced Arthritis"

_ijms, 2023, doi:10.3390/ijms24098223_

Round 1
Reviewer 1 Report
This work describes that insertion of BMP-2 into AMMs generated by gene editing could be a promising alternative strategy to increase anti-arthritic potential.
In my opinion, the paper is very interesting and well written. The methodology is described in accordance with scientifically accepted standards. The description is detailed enough to allow the reader to review the research. The results are described in detail. In the discussion, reference is made to the results of other authors and appropriate conclusions are drawn. However, in my estimation, more pro-inflammatory cytokines and expression of critical inflammatory genes are lacking to say that anti-inflammatory properties have been demonstrated. COX-2 and the NFk-B factor are mentioned in other studies but not identified in the submitted manuscript would also be an excellent indicator of an anti-inflammatory state.
Author Response
Reviewer #1
This work describes that insertion of BMP-2 into AMMs generated by gene editing could be a promising alternative strategy to increase anti-arthritic potential.
In my opinion, the paper is very interesting and well written. The methodology is described in accordance with scientifically accepted standards. The description is detailed enough to allow the reader to review the research. The results are described in detail. In the discussion, reference is made to the results of other authors and appropriate conclusions are drawn.
However, in my estimation, more pro-inflammatory cytokines and expression of critical inflammatory genes are lacking to say that anti-inflammatory properties have been demonstrated. COX-2 and the NFk-B factor are mentioned in other studies but not identified in the submitted manuscript would also be an excellent indicator of an anti-inflammatory state.
Response
We added the data regarding the expression level of COX-2 and NFk-B in the Figure 6.
Thanks for the valuable comments.

Reviewer 2 Report
The manuscript by Chae et al. examined the role of genome-edited BMP-2 overexpressing Amniotic Mesenchymal Stem Cells (AMMs) collagen-induced arthritis (CIA) mouse models. The BMP-2 was overexpressed into AMMs using Transcription Activator-Like Effector Nucleases (TALENs) system. Cell sorting and qRT-PCR were used to characterize BMP-2 overexpressing AMMs.
Further, in co-culture of AMM/B with tumor necrosis factor (TNF)-α treated synovial fibroblasts significantly decreased interleukin (IL) production. Also, the therapeutic potential of AMM/B was evaluated using the CIA mouse model. Injection of AMM/B attenuated CIA progression and inhibited T-helper (Th)17 cell activation in CIA mice. Overall this is a nicely conducted study where the data support conclusions well.
Here are some concerns:
- In Fig. 3C, why injection of AMM alone without BMP-2 overexpression is improving the clinical score? Is it because of the basal level of secretion of BMP-2?
- Would it be informative to know the amount of BMP-2 protein secretion upon overexpressing the Bmp-2 genes compared to WT-AMM?
- Have the authors checked the active BMP signaling by looking at the phosphorylation level of Smad 1/5/8?
- Figure 5, in histological analysis, which joints were analyzed? Were there any joint/site-specific differences in terms of the phenotypes?
- What could be a possible mechanism for reducing the interleukins/cytokines? Are these cells autonomous or cell non-autonomuse effects of BMP-2?
Author Response
Reviewer #2
The manuscript by Chae et al. examined the role of genome-edited BMP-2 overexpressing Amniotic Mesenchymal Stem Cells (AMMs) collagen-induced arthritis (CIA) mouse models. The BMP-2 was overexpressed into AMMs using Transcription Activator-Like Effector Nucleases (TALENs) system. Cell sorting and qRT-PCR were used to characterize BMP-2 overexpressing AMMs.
Further, in co-culture of AMM/B with tumor necrosis factor (TNF)-α treated synovial fibroblasts significantly decreased interleukin (IL) production. Also, the therapeutic potential of AMM/B was evaluated using the CIA mouse model. Injection of AMM/B attenuated CIA progression and inhibited T-helper (Th)17 cell activation in CIA mice. Overall this is a nicely conducted study where the data support conclusions well.
Here are some concerns:
- In Fig. 3C, why injection of AMM alone without BMP-2 overexpression is improving the clinical score? Is it because of the basal level of secretion of BMP-2?
Response
AMMs also have been reported to have some therapeutic potential such as anti-inflammatory. Those were described in the discussion section. AMMs possess anti-inflammatory, chondroprotective, and regenerative properties. [1, 2].
2.Would it be informative to know the amount of BMP-2 protein secretion upon overexpressing the Bmp-2 genes compared to WT-AMM?
Response
We added the protein level of BMP-2 in the Figure 1E.
3.Have the authors checked the active BMP signaling by looking at the phosphorylation level of Smad 1/5/8?
Response
We have not checked the active BMP signaling. However, we found that co-culture with AMM/B significantly decreased IL-34 levels compared to co-culturing with AMMs in synovial fibroblast. These data indicate the existence of active BMP signaling factor derived from AMM/B.
4.Figure 5, in histological analysis, which joints were analyzed? Were there any joint/site-specific differences in terms of the phenotypes?
Response
We added the location of joint in the Figure legnend of Figure 5.
5.What could be a possible mechanism for reducing the interleukins/cytokines? Are these cells autonomous or cell non-autonomuse effects of BMP-2?
Response
BMP-2 might have a role in reducing cytokine levels, potentially by directly affecting the cells producing the cytokines (autonomous effect). Further research is needed to fully understand the autonomous or non-autonomous effects of BMP-2 on cytokine production. This could involve studying the BMP-2 signaling pathways and its interactions with specific immune cells, as well as examining the effects of BMP-2 on other cytokines and in different experimental contexts.
We added this limitation in the Discussion section.
References.
- Topoluk, N.; Steckbeck, K.; Siatkowski, S.; Burnikel, B.; Tokish, J.; Mercuri, J., Amniotic mesenchymal stem cells mitigate osteoarthritis progression in a synovial macrophage-mediated in vitro explant coculture model. Journal of tissue engineering and regenerative medicine 2018, 12, (4), 1097-1110.
- Ragni, E.; Papait, A.; Perucca Orfei, C.; Silini, A. R.; Colombini, A.; Vigano, M.; Libonati, F.; Parolini, O.; de Girolamo, L., Amniotic membrane-mesenchymal stromal cells secreted factors and extracellular vesicle-miRNAs: Anti-inflammatory and regenerative features for musculoskeletal tissues. Stem cells translational medicine 2021, 10, (7), 1044-1062.

Round 2
Reviewer 2 Report
In the revised version of the manuscript, the authors have addressed the concerns I raised satisfactorily.
Minor editing is required.